# EmoNet-Voice: A Large-Scale Synthetic Benchmark for Fine-Grained Speech Emotion

## Abstract

Speech emotion recognition (SER) systems are constrained by existing datasets that typically cover only 6-10 basic emotions, lack scale and diversity, and face ethical challenges when collecting sensitive emotional states. We introduce EMONET-VOICE, a comprehensive resource addressing these limitations through two components: (1) EmoNet-Voice Big, a 5,000-hour multilingual pre-training dataset spanning 40 fine-grained emotion categories across 11 voices and 4 languages, and (2) EmoNet-Voice Bench, a rigorously validated benchmark of 4,7k samples with unanimous expert consensus on emotion presence and intensity levels. Using state-of-the-art synthetic voice generation, our privacy-preserving approach enables ethical inclusion of sensitive emotions (e.g., pain, shame) while maintaining controlled experimental conditions. Each sample underwent validation by three psychology experts. We demonstrate that our Empathic Insight models trained on our synthetic data achieve strong real-world dataset generalization, as tested on EmoDB and RAVDESS. Furthermore, our comprehensive evaluation reveals that while high-arousal emotions (e.g., anger: 95% accuracy) are readily detected, the benchmark successfully exposes the difficulty of distinguishing perceptually similar emotions (e.g., sadness vs. distress: 63% discrimination), providing quantifiable metrics for advancing nuanced emotion AI. EMONET-VOICE establishes a new paradigm for large-scale, ethically-sourced, fine-grained SER research.

## 1 Introduction

Synthetic speech technology has reached unprecedented fidelity, with state-of-the-art text-to-speech (TTS) and audio generation models, e.g., GPT-4 OmniAudio (OpenAI, 2024), achieving near-human fidelity. These advancements significantly enhance human-computer interaction (HCI), enabling virtual assistants to convey appropriate emotional qualities across diverse contexts (Kirk et al., 2025). However, this advancement remains asymmetric: while machines can to some extent effectively *synthesize* convincing affective speech, they still struggle to *recognize* the nuanced, context-dependent emotional information humans naturally convey (Lee & Gomez, 2021; Schuller, 2018), a critical capability for truly conversational AI.

Despite steady progress in speech emotion recognition (SER) through deep architectures and self-supervised representations, evaluation remains constrained by datasets predominantly built around a limited set of "basic" emotions (Ekman, 1992; Zhao & Kumar, 2022). Established benchmarks such as IEMOCAP (Busso et al., 2008), RAVDESS (Livingstone & Russo, 2018), and CREMA-D (Cao et al., 2014) have been invaluable for the field but exhibit three fundamental limitations: (i) **Insufficient Granularity.** Coarse taxonomies fail to capture subtle or compound emotional states (e.g., *bittersweet*, *embarrassment*, *envy*) that are essential for naturalistic interaction (Cowen et al., 2019). (ii) **Limited Representativeness.** Current datasets predominantly consist of studio-quality acted speech, lacking linguistic diversity and omitting sensitive emotional states due to privacy constraints (Lorenzo-Trueba et al., 2017; Schuller et al., 2013). (iii) **Restricted Scalability.** Licensing restrictions, privacy concerns, and annotation costs severely limit dataset size, impeding the data-intensive training regimes required by modern deep learning approaches (Zhang et al., 2020; Poria et al., 2020), specifically for open-source and -science.

These limitations are compounded by evolving views in affective science, which frame emotions as context-dependent constructions rather than fixed categories (Barrett, 2017). Dimensional models

Table 1: **Comparison of SER datasets**. Key aspects include licensing, size, emotional range, speaker diversity, synthetic origin, and multilingual support. Open license means CC-BY 4.0 or equivalent; var. means varies across pooled corpora.

| Dataset | Open Licence | Size (#Utts/Hours) | #Emo. | #Spk. | Synth. | Multilin. |
|---|---|---|---|---|---|---|
| IEMOCAP (Busso et al., 2008) | ✗ | 10k / ∼12h | 9 | 10 (5M/5F) | ✗ | ✗ |
| RAVDESS (Livingstone & Russo, 2018) | ✓ | 1.4k / ∼1h | 8 | 24 (12M/12F) | ✗ | ✗ |
| SAVEE (Jackson & Haq, 2014) | ✗ | 480 / <1h | 7 | 4 (Male) | ✗ | ✗ |
| EmoDB (Burkhardt et al., 2005) | ✗ | 535 / <1h | 7 | 10 (5M/5F) | ✗ | ✗ |
| CREMA-D (Cao et al., 2014) | ✓ | 7.4k / ∼6h | 6 | 91 (48M/43F) | ✗ | ✗ |
| SERAB (Scheidwasser-Clow et al., 2021) | ✗ | 9 corpora / var. | 6 | var. | ✗ | ✓ |
| EmoBox (Ma et al., 2024) | ✗ | 32 corpora / var. | ≤8 | var. | ✗ | ✓ |
| SER Evals (Osman et al., 2024) | ✗ | 18 corpora / var. | ≤8 | var. | ✗ | ✓ |
| BERSt (Tuttösí et al., 2025) | ✓ | ∼4h | 6 | 98 | ✗ | ✗ |
| **EMONET-VOICE BIG** | ✓ | >1M / >4,500h | 40 | **11 (Synth)** | ✓ | ✓ |
| **EMONET-VOICE BENCH** | ✓ | ∼12k / 35.8h | 40 | **11 (Synth)** | ✓ | ✓ |

such as the valence–arousal circumplex (Russell, 1980) reinforce the need for richer datasets and modeling strategies beyond discrete classification (Schuhmann et al., 2025).

To address these challenges, we introduce two complementary datasets. First, **EMONET-VOICE BIG**, a foundational dataset for pretraining models on SER. It is a comprehensive synthetic voice corpus of 5,000 hours in four languages (English, German, Spanish, French), featuring 11 distinct voices with different (gender) identities and a fine-grained taxonomy of 40 emotion categories. As such, it provides an open, privacy-compliant foundation for emotional TTS research and multilingual speech analysis at scale. Second, from this corpus we curate **EMONET-VOICE BENCH**, comprising 12,600 audio clips annotated by psychology experts using a strict consensus protocol that evaluates *both* the presence *and* intensity of each target emotion across our 40-category emotion taxonomy. This approach yields a high-quality, multilingual benchmark for fine-grained SER while circumventing the privacy barriers that inhibit the collection of authentic sensitive vocal expressions.

Building on our pretraining dataset, we develop **EMPATHICINSIGHT-VOICE** (SMALL and LARGE), novel SER models that achieve state-of-the-art performance in fine-grained SER while demonstrating strong alignment with human expert judgments. Through comprehensive evaluation across the concurrent SER model landscape, we reveal critical insights into current SER capabilities, including systematic patterns in which emotions prove more challenging to recognize (e.g., low-arousal states like concentration versus high-arousal emotions like anger). Finally, we demonstrate strong real-to-sim generalization of our models, trained on our synthetic data, across two real-world datasets.

In summary, our contributions are: **(1)** We build **EMONET-VOICE BIG**, a pretraining, open-access, 5,000-hour multilingual synthetic speech corpus featuring 11 distinct synthetic voices across 4 languages and 40 emotion categories. **(2)** We introduce **EMONET-VOICE BENCH**, a meticulously curated and expert-verified benchmark dataset of 13k high-quality audio samples for fine-grained SER, featuring 40 emotions with 3 intensities. **(3)** We build **EMPATHICINSIGHT-VOICE** (Small and Large), novel SER models designed for nuanced emotion estimation. **(4)** We conduct comprehensive evaluations on our benchmark, providing critical insights into current SER capabilities and limitations.

## 2 RELATED WORK

**Current SER research operates on a constrained empirical foundation.** The field currently relies on a few acted corpora recorded in controlled studios—IEMOCAP (12h, 9 emotions) (Busso et al., 2008), RAVDESS (1h, 8 emotions) (Livingstone & Russo, 2018), SAVEE (0.8h, 7 emotions) (Jackson & Haq, 2014), the German EMODB (Burkhardt et al., 2005), and the multi-ethnic CREMA-D (Cao et al., 2014). While these offer clean labels and high acoustic quality, they share four key weaknesses. First, they use restrictive taxonomies—typically six basic emotions from Ekman (1992)—omitting compound or socially nuanced states such as envy, or contemplation (Plutchik, 2001; Cowen et al., 2019). Second, acted prosody exaggerates cues, limiting generalization to spontaneous speech (Lorenzo-Trueba et al., 2017). Third, privacy and ethics hinder collection of intimate or

stigmatizing emotions (e.g., shame, desire) (Schuller et al., 2013). Fourth, scale and linguistic diversity are limited: most corpora have <100 speakers, few hours of audio, and focus on English. Recent expansions include multilingual sets like EMOREACT, which broaden emotion categoriesbut still retain one language, English, (Nojavan & Soleymani, 2021). Aggregation benchmarks like SERAB (nine corpora, six languages) (Scheidwasser-Clow et al., 2021), EMOBOX (32 datasets, 14 languages) (Ma et al., 2024), SER EVALS (18 minority-language corpora) (Osman et al., 2024), and BERST (4h shouted speech, 98 actors, 19 smartphone positions) (Tuttösí et al., 2025) extend coverage but inherit core limits: acted/scripted speech, narrow taxonomies ($\leq 8$ emotions), and no expert-validated intensities or sensitive states. These datasets, summarized in Tab. 1, reveal a clear gap: they are often restricted by licensing, limited scale (hours and utterances), narrow emotion range (typically $\leq 8$), rely on actors limiting privacy-sensitive emotions, and lack multilingual scope. EMONET-VOICE BIG and BENCH address these by providing a large-scale, openly licensed, synthetic, multilingual corpora with a 40-emotion taxonomy.

**Taxonomic limitations exacerbate data-scarcity and theoretical gaps.** Modern affective science models emotions as context-dependent and graded rather than discrete (Barrett, 2017; Lindquist, 2013). Dimensional (valence–arousal–dominance) and multi-label schemes (Russell, 1980; Zhang et al., 2020) better capture blended affect, yet almost all benchmarks still assign a *single discrete label* per clip. When intensity annotations exist, they typically rely on crowdsourcing and show low agreement (Kajiwara et al., 2021; Stappen et al., 2021). Consequently, the community lacks benchmarks that reflect contemporary understanding of emotion as multidimensional and graded, particularly for sensitive affective states that cannot be ethically collected from human participants. Expert-validated intensity annotations across multidimensional affective spaces are missing from existing benchmarks, and we fill this critical gap by contributing EMONET-VOICE BENCH with 12,600 carefully chosen clips whose emotional *presence* and *intensity* we had annotated by psychology experts, yielding a high-agreement subset. We overcome previous taxonomic, scale, and ethical limitations by combining multilingual coverage, a 40-category taxonomy grounded in contemporary affective science (Cowen et al., 2020; Barrett, 2017), and privacy-preserving synthetic speech generation, offering the first benchmark that provides *expert* ratings across a multidimensional affective space.

## 3   THE EMONET-VOICE SUITE: DATASET CONSTRUCTION

This section covers building the EMONET-VOICE resources: the emotion taxonomy, the large-scale pre-training dataset EMONET-VOICE BIG, the expert-validated EMONET-VOICE BENCH, and finally the EMPATHICINSIGHT-VOICE models that set a new SER standard.

### 3.1   EMONET-VOICE EMOTION TAXONOMY

For EMONET-VOICE, we adopt the comprehensive 40-category emotion taxonomy originally developed for EMONET-FACE (Schuhmann et al., 2025). The taxonomy includes a diverse set of categories spanning positive emotions (e.g., *Elation*, *Contentment*, *Affection*, *Awe*), negative emotions (e.g., *Distress*, *Sadness*, *Bitterness*, *Contempt*), cognitive states (e.g., *Concentration*, *Confusion*, *Doubt*), physical states (e.g., *Pain*, *Fatigue*), and socially mediated emotions (e.g., *Embarrassment*, *Shame*, *Pride*, *Teasing*). This fine-grained structure enables the evaluation of models beyond binary or basic categorical classification. The full set of 40 emotion categories and their descriptive terms can be found in App.A.1. A comprehensive description of the methodology used to construct the taxonomy, including literature-based extraction and expert-guided refinement, is provided in App.A.4.

### 3.2   EMONET-VOICE BIG: BUILDING A LARGE-SCALE SYNTHETIC SER DATASET

The foundational dataset, EMONET-VOICE BIG, consists of emotionally expressive speech samples synthesized using GPT-4 OmniAudio. An overview of EMONET-VOICE BIG's scale and language distribution is provided in Table 2. Our prompting strategy cast the model as an actor auditioning for a film, tasked with performing texts designed to evoke one of 40 emotion categories (from the taxonomy in Section 3.1). Key prompt elements included directives for strong emotional expression from the outset and naturalistic human speech patterns (e.g., varied rhythm, volume, tone, and appropriate vocal bursts). This aimed to ensure perceptible emotional content and avoid monotonous delivery. Audio was generated as 3- to 30-second, 24kHz WAV files, utilizing 11 synthetic voices (6

Table 2: Overview of EMONET-VOICE BIG

| Category | Hours |
|---|---|
| **Playtime by Language** | |
| English (en) | 2,156 |
| German (de) | 716 |
| Spanish (es) | 888 |
| French (fr) | 881 |
| Acting Chal. (en+de) | 111 |
| total | **4,752** |
| **English Accent Distribution** | |
| Louisiana | 133 |
| Valley Girl | 159 |
| British | 132 |
| Chinese | 126 |
| French | 140 |
| German | 135 |
| Indian | 129 |
| Italian | 134 |
| Mexican | 131 |
| Russian | 134 |
| Spanish | 132 |
| Texan | 131 |
| Vulgar Street | 149 |
| No accent specified | 391 |

Table 3: Overview of EMONET-VOICE BENCH

| Category | Value |
|---|---|
| **Number of Clips** | |
| English (en) | 6,156 (48.9%) |
| German (de) | 1,886 (15.0%) |
| Spanish (es) | 2,193 (17.4%) |
| French (fr) | 2,365 (18.8%) |
| **Total Clips** | **12,600** |
| **Avg. Clip Duration** | **10.36 s** |
| **Total Playtime** | **36.26 h** |

Table 4: Number of voice audios annotated by human experts across batches for EMONET-VOICE BENCH. Mainly samples with at least positive weak agreement (emotion weakly / strongly present annotated by two human experts) were used in a next batch.

| Batch | Unique Human Annotators | Annotated Voice Audios |
|---|---|---|
| 1 | 2 | 4,538 |
| 2 | 3 | 7,719 |
| 3 | 4 | 343 |

female/5 male) across English, German, French, and Spanish to build a diverse multilingual corpus. We show more details on the prompting template and methodology, e.g., instruction sensitivity and language-specific adaptations for vocal burst generation, in the Supplement.

### 3.3 EMONET-VOICE BENCH: A HUMAN EXPERT BENCHMARK FOR SER

From EMONET-VOICE BENCH, we created a subset of 12,600 unique audio files annotated for emotion by human experts on a three-point annotation scale, summarized in Table 3. We depict the annotation platform for our human experts in Appendix Figures 2 and 3. The dataset features 11 distinct synthetic voices (6 female and 5 male) across four languages: English (48.9%), German (15.0%), Spanish (17.4%), and French (18.8%). The average clip duration is 10.36 seconds, resulting in a total playtime of 36.26 hours.

Table 4 summarizes our annotation procedure. Ensuring the quality and reliability of the emotion annotations was a central priority in constructing the EMONET-VOICE BENCH. We recruited a team of six human experts with at least a Bachelor's degree in Psychology to serve as benchmark annotators, thereby guaranteeing familiarity with emotional theory and terminology. In total, 33,605 single-emotion labels across 12,600 unique audio samples were contributed — some samples ultimately received more than three annotations. Each audio clip was first labeled independently by two experts who were presented with the audio alongside one specific target emotion category from our taxonomy in addition to a three-point scale: 0 indicating the emotion was not perceived, 1 indicating it was mildly present at low intensity, and 2 indicating it was intensely present and clearly perceptible. If both human experts agreed that the emotion was present (either "weakly present" or "strongly present"), the clip was sent to a third expert for confirmation. Additionally, we randomly selected a subset of clips to receive a third or even a fourth annotation regardless of whether the first two annotators agreed. To reduce potential gender biases in emotional perception, each group assigned per snippet was balanced in gender composition. Importantly, annotators performed their assessments independently and were blinded to the ratings of others.

Figure 1 illustrates inter-annotator agreement patterns across emotion categories, showing the distribution of full agreement, partial agreement, and disagreement for each emotion-audio pair. The numbers alongside each bar indicate total instances and rating distributions across multiple annotators. The analysis reveals clear consensus patterns: emotions like *concentration* and *bitterness* achieve strong expert agreement, while others such as *numbness* and *awe* show notable disagreement even

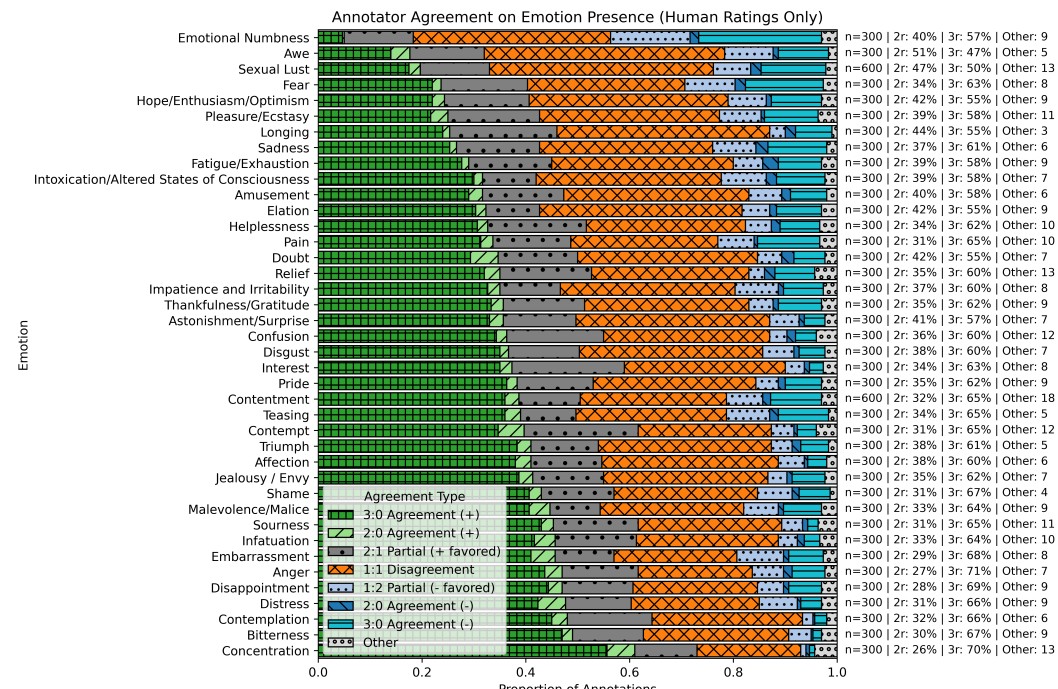

Figure 1: Expert annotator agreement on perceived emotions. Stacked bars show the proportion of audio-emotion instances by agreement type, from unanimous agreement on presence (e.g., '3:0 (+)') to disagreement ('1:1') and unanimous agreement on absence ('3:0 (-)'). Numbers to the right indicate total instances (n) per emotion and the distribution of raters (%2r, %3r). The patterns reveal high consensus for acoustically salient emotions like *concentration* but significant ambiguity for nuanced states like *awe*, underscoring the challenge of fine-grained SER.

among psychology professionals. The overall inter-rater reliability measured by Cronbach's $\alpha$ is 0.14 (95% CI [0.12, 0.15]), with per-emotion values detailed in Appendix Table 10. While this low $\alpha$ might initially suggest poor reliability, it actually reflects the inherent complexity of fine-grained emotion perception rather than annotation deficiencies. Unlike simpler emotion taxonomies, our 40-category framework captures subtle distinctions that legitimately evoke different interpretations among experts. These patterns demonstrate that while human agreement is robust for many emotions, certain categories naturally elicit diverse interpretations—underscoring the nuanced nature of affective expression in speech. Rather than indicating weak annotation quality, this variability highlights EMONET-VOICE's sensitivity to the inherent complexity of emotional perception. Our annotations thus capture both the challenges and opportunities in modeling authentic emotional diversity at scale.

### 3.4 EMPATHICINSIGHT-VOICE: TRAINING STATE-OF-THE-ART SER MODELS

Leveraging our datasets, we further contribute by developing novel state-of-the-art SER models.

In first linear probing experiments, and previous works (Li et al., 2023; Dutta & Ganapathy, 2024), we observe that the off-the-shelf Whisper encoders (Radford et al., 2023) are not capable of reflecting on emotions. Specifically, at a fine-grained level, existing TTS models fail to recognize emotions effectively, as we will discuss later. To address this limitation, we continually pre-trained Whisper encoders as the backbone of our EMPATHICINSIGHT-VOICE. Specifically, we leverage EMONET-VOICE BIG as a pretraining dataset and train emotion-experts in two stages. We base our experiments on Whisper-Small to optimize for the performance-efficiency tradeoff.

In the first stage, the Whisper encoder is trained on a combination of EMONET-VOICE BIG and another 4,500 hours of public emotion-related content[1] to develop general emotional acoustic rep-

---

[1]https://huggingface.co/datasets/mitermix/audiosnippets

Table 5: Performance comparison of audio language models on the EMONET-VOICE BENCH. Models are evaluated against human emotion ratings using correlation metrics (Spearman and Pearson $r$, higher is better) and error metrics (MAE and RMSE, lower is better). Our EMPATHICINSIGHT-VOICE models demonstrate superior performance across all metrics, with LARGE achieving the highest Pearson correlation and lowest error and refusal rates. Refusal rates indicate the percentage of samples where models declined to provide emotion assessments. Best scores in bold.

| Model | Refusal ($\downarrow$) | Spearman ($\uparrow$) | Pearson ($\uparrow$) | MAE ($\downarrow$) | RMSE ($\downarrow$) |
|---|---|---|---|---|---|
| Gemini 2.0 Flash | 0.01% | 0.355 | 0.350 | 3.608 | 4.453 |
| Gemini 2.5 Pro | **0.00**% | 0.417 | 0.416 | 3.008 | 3.785 |
| GPT-4o Mini Audio Preview | 2.26% | 0.326 | 0.327 | 3.320 | 4.124 |
| GPT-4o Audio Preview 2024-12-17 | 27.59% | 0.337 | 0.336 | 3.432 | 4.247 |
| Hume Voice | 39.16% | 0.274 | 0.231 | 4.744 | 5.474 |
| EMPATHICINSIGHT-VOICE SMALL | **0.00**% | **0.418** | 0.414 | 2.997 | 3.757 |
| EMPATHICINSIGHT-VOICE LARGE | **0.00**% | 0.415 | **0.421** | **2.995** | **3.756** |

resentations. This data was annotated using an iterative process with Gemini Flash 2.0 to obtain emotion scores (0–4 scale) for all audio snippets. In the second stage, we freeze the Whisper encoder and train MLP expert heads—one per emotion dimension—on top of the fixed encoder embeddings. This way, each MLP receives the full voice audio sequence from the Whisper encoder as sequence flattened token embeddings and then regresses a single emotion intensity score. We propose two model sizes to accommodate different performance requirements, namely EMPATHICINSIGHT-VOICE SMALL with 74M paremeter MLP heads and EMPATHICINSIGHT-VOICE LARGE with 148M paremeter MLP heads. We optimize them using mean absolute error (MAE) on the Gemini Flash 2.0–generated emotion scores. Through this two-stage fine-tuning and dedicated MLP ensemble, EMPATHICINSIGHT-VOICE effectively captures and predicts fine-grained emotional content from speech with high human alignment, as we demonstrate in the following. More details in App. A.2.

# 4 EXPERIMENTS: DO THEY HEAR WHAT WE HEAR?

In this section, we evaluate current SER models on our novel benchmark. Before that, we start by introducing our experimental setup.

**Experimental Setup.** EMONET-VOICE BENCH assesses a model's proficiency in discerning emotional intensity from audio. To facilitate a nuanced comparison across models, many of which output continuous scores, our primary evaluation employs metrics suited for regression and correlation analysis on a common scale. The 3-level intensity human judgments (0: Not Present, 1: Mildly Present, 2: Intensely Present) are mapped to a 0-10 scale for this evaluation, becoming 0, 5, and 10, respectively. Model predictions are likewise generated or normalized to this 0-10 continuous scale.

We benchmarked general-purpose multimodal models (e.g., Gemini, GPT-4o) via zero-shot prompting, as well as specialized speech models (e.g., Hume Voice). Hume Voice was subject to constraints on input length ($\leq$5s) and taxonomy coverage. Initial experiments with Whisper failed, due to a general lack of emotion understanding, which led to our development of EMPATHICINSIGHT-VOICE, which pair continually pre-trained Whisper encoders with MLP regressors on our EMONET-VOICE dataset.

We report four key metrics: *Mean Absolute Error (MAE)* and *Root Mean Squared Error (RMSE)* to quantify the average magnitude and larger deviations of prediction error on this 0-10 scale. Additionally, *Pearson Correlation (Pearson r)* and *Spearman Rank Correlation (Spearman r)* are used to assess the linear and monotonic agreement, respectively, between model-predicted intensities and human judgments. These metrics collectively provide a comprehensive view of how well models capture both the absolute values and the relative ordering of perceived emotional intensities.

## 4.1 EVALUATING SPEECH EMOTION RECOGNITION MODELS

Table 5 presents performance across seven models on the unseen EMONET-VOICE BENCH, revealing clear performance tiers. Our EMPATHICINSIGHT-VOICE models achieve state-of-the-art results, with

Table 6: Spearman's $\rho$ by emotion for audio models. Emotions are sorted by average model performance, with best values in bold, runner-up underlined, and color-coded by correlation (gradient from red $= -1$ to blue $= 1$, NaN in gray). Key patterns: (i) There is generally strong alignment with humans for high-arousal emotions like teasing. (ii) Our EMPATHICINSIGHT-VOICE models consistently outperform or place second across emotions. (iii) Some commercial models show systematic refusal (NaNs) for sensitive emotions (e.g., sexual content). (iv) Performance drops for low-arousal emotions (e.g., concentration). (v) Even SOTA models struggle with complex cognitive-emotional states (e.g., contemplation), suggesting general limits to detect less physiological emotions.

| emotion | GPT-4o Mini Audio | GPT-4o Audio | Hume Voice | Gemini 2.0 Flash | Gemini 2.5 Pro | EMPATHICINSIGHT-VOICE SMALL (ours) | EMPATHICINSIGHT-VOICE LARGE (ours) | avg. |
|---|---|---|---|---|---|---|---|---|
| Teasing | 0.569 | 0.636 | NaN | 0.556 | 0.626 | 0.649 | **0.662** | 0.617 |
| Embarrassment | 0.550 | 0.637 | 0.416 | 0.529 | 0.618 | 0.669 | **0.678** | 0.585 |
| Anger | 0.496 | 0.555 | 0.418 | 0.526 | **0.602** | 0.578 | 0.577 | 0.536 |
| Impatience and Irritability | 0.455 | 0.471 | NaN | 0.448 | 0.504 | 0.554 | **0.570** | 0.500 |
| Malevolence/Malice | 0.345 | NaN | NaN | 0.333 | 0.529 | 0.562 | **0.615** | 0.477 |
| Shame | 0.437 | 0.393 | 0.441 | 0.419 | 0.516 | 0.552 | **0.558** | 0.474 |
| Sadness | 0.470 | 0.404 | 0.357 | 0.466 | **0.529** | 0.483 | 0.521 | 0.461 |
| Helplessness | 0.347 | 0.375 | NaN | 0.462 | 0.483 | 0.536 | 0.535 | 0.457 |
| Astonishment/Surprise | **0.487** | NaN | NaN | 0.454 | 0.459 | 0.451 | 0.428 | 0.456 |
| Pleasure/Ecstasy | 0.364 | NaN | NaN | 0.342 | 0.462 | 0.538 | 0.529 | 0.447 |
| Disgust | 0.421 | **0.493** | 0.330 | 0.419 | 0.483 | 0.419 | 0.460 | 0.432 |
| Contempt | 0.412 | 0.433 | 0.324 | 0.407 | 0.466 | 0.478 | 0.469 | 0.427 |
| Fear | 0.355 | 0.367 | 0.437 | 0.353 | 0.441 | **0.470** | 0.458 | 0.411 |
| Amusement | 0.412 | 0.362 | 0.380 | 0.355 | 0.432 | 0.454 | **0.462** | 0.408 |
| Relief | 0.317 | 0.361 | 0.398 | 0.349 | 0.463 | 0.462 | **0.501** | 0.407 |
| Pain | 0.365 | 0.345 | 0.370 | 0.386 | 0.413 | 0.472 | **0.474** | 0.404 |
| Jealousy/ Envy | 0.334 | 0.361 | 0.264 | 0.425 | **0.487** | 0.469 | 0.471 | 0.402 |
| Elation | 0.390 | 0.330 | 0.313 | 0.344 | 0.466 | 0.475 | **0.487** | 0.401 |
| Pride | 0.348 | 0.308 | 0.259 | 0.415 | 0.482 | **0.484** | 0.474 | 0.396 |
| Confusion | 0.379 | 0.339 | 0.331 | 0.358 | **0.451** | 0.423 | 0.451 | 0.390 |
| Disappointment | 0.301 | **0.466** | 0.249 | 0.370 | 0.426 | 0.432 | 0.461 | 0.386 |
| Doubt | 0.379 | 0.347 | 0.241 | 0.403 | 0.402 | 0.459 | **0.463** | 0.385 |
| Triumph | 0.333 | 0.279 | 0.216 | 0.370 | **0.482** | 0.460 | 0.455 | 0.371 |
| Infatuation | 0.315 | 0.317 | NaN | 0.354 | **0.413** | 0.392 | 0.408 | 0.367 |
| Bitterness | 0.330 | 0.324 | NaN | 0.286 | 0.360 | **0.411** | 0.404 | 0.352 |
| Fatigue/Exhaustion | 0.221 | NaN | NaN | 0.297 | 0.400 | **0.455** | 0.384 | 0.351 |
| Thankfulness/Gratitude | 0.297 | NaN | NaN | 0.281 | **0.418** | 0.358 | 0.379 | 0.347 |
| Intoxication/Altered States of Consciousness | 0.198 | NaN | NaN | 0.269 | 0.241 | 0.486 | **0.487** | 0.336 |
| Distress | 0.374 | 0.369 | -0.138 | 0.375 | **0.450** | 0.432 | 0.430 | 0.327 |
| Sexual Lust | 0.203 | 0.279 | NaN | 0.356 | **0.450** | 0.332 | 0.334 | 0.326 |
| Affection | 0.310 | **0.390** | 0.182 | 0.330 | 0.349 | 0.359 | 0.356 | 0.325 |
| Longing | 0.289 | 0.330 | 0.214 | 0.326 | 0.348 | 0.365 | 0.350 | 0.317 |
| Awe | 0.298 | 0.276 | 0.058 | 0.314 | 0.314 | 0.329 | **0.332** | 0.275 |
| Hope/Enthusiasm/Optimism | 0.250 | NaN | NaN | 0.175 | 0.203 | **0.345** | 0.343 | 0.263 |
| Sourness | 0.158 | 0.180 | NaN | 0.250 | 0.303 | **0.331** | 0.323 | 0.258 |
| Interest | 0.161 | 0.169 | 0.119 | 0.148 | 0.287 | **0.351** | 0.315 | 0.221 |
| Contemplation | 0.187 | 0.128 | 0.177 | 0.252 | **0.282** | 0.263 | 0.247 | 0.219 |
| Contentment | -0.044 | -0.019 | 0.195 | 0.140 | 0.224 | 0.231 | **0.330** | 0.151 |
| Emotional Numbness | 0.139 | 0.092 | NaN | 0.099 | 0.125 | 0.139 | **0.145** | 0.123 |
| Concentration | 0.085 | 0.019 | **0.262** | 0.186 | 0.151 | 0.055 | 0.068 | 0.118 |

EMPATHICINSIGHT-VOICE LARGE obtaining the highest Pearson correlation (0.421) and lowest error rates (MAE: 2.995, RMSE: 3.756). EMPATHICINSIGHT-VOICE SMALL demonstrates competitive performance with the highest Spearman correlation (0.418). Gemini 2.5 Pro emerges as the strongest foundation model competitor (Pearson r: 0.416, Spearman r: 0.417), while other commercial models show significantly lower correlations and higher error and refusal rates. This shows that current audio models show decent alignment with human expert ratings on SER. Notably, refusal rates vary drastically across models. While EMPATHICINSIGHT-VOICE models and Gemini variants process all samples (0-0.01% refusal), GPT-4o Audio Preview refuses 27.59% of samples, and Hume Voice refuses 39.16%—reflecting safety constraints around sensitive emotional content, such as intoxication and pleasure/ecstasy. Overall, this indicates that our specialized AI models can, to some extent, "hear" what humans hear and demonstrate reasonable alignment with human emotion ratings, while several (general-purpose) models struggle in this task. Yet, this recognition capability proves more complex than initially apparent, as we will explore further next.

**Emotion-Specific Performance Patterns.** Per-emotion analysis in Table 6 reveals clear performance hierarchies. High-arousal emotions prove most detectable across all models: *teasing* (average Spearman r: 0.617), *embarrassment* (0.585), and *anger* (0.536) show strong human-model alignment, suggesting these acoustic signatures are most reliably encoded in prosody. Conversely, performance

drops dramatically for subtle, low-arousal states like *concentration* (0.118) and *emotional numbness* (0.123), highlighting fundamental limitations in detecting nuanced emotional states from audio alone.

Moreover, the table reveals systematic differences in emotion detection across our 40-category taxonomy. It demonstrates that EMPATHICINSIGHT-VOICE models consistently outperform competitors across most emotions, particularly excelling in complex states often missed by other systems. For instance, EMPATHICINSIGHT-VOICE achieves superior performance on challenging emotions like *intoxication* (where EMPATHICINSIGHT-VOICE scores 0.48 compared to 0.269 by the runner-up and many commercial models often completely refuse assessment), and similar for *malevolence*—emotions that require nuanced prosodic understanding.

**Commercial Model Limitations.**   Commercial models exhibit systematic refusal patterns for sensitive content, with GPT-4o Audio and Hume Voice showing nearly identical NaN patterns for emotions like *sexual content* and *intoxication*—indicating shared (safety) constraints. This creates evaluation gaps precisely where human emotional complexity is especially relevant for applications. Even state-of-the-art models struggle with complex cognitive-emotional states (*contemplation*, *interest*, *contentment*), suggesting current architectures may be fundamentally limited to more physiologically manifest emotions rather than subtle internal states.

## 4.2 CROSS-DATASET GENERALIZATION TO REAL-WORLD DATA

A key question for any synthetic dataset is whether models trained on it can generalize to real-world data. To assess this synthetic-to-real transfer capability, we evaluated our EMPATHICINSIGHT-VOICE LARGE model, trained exclusively on EMONET-VOICE BIG, on two widely-used human-acted SER benchmarks: EmoDB (Burkhardt et al., 2005) and RAVDESS (Livingstone & Russo, 2018). A significant challenge in this evaluation is the "semantic gap" between our 40 fine-grained emotion categories and the 7-8 coarse categories used in these benchmarks. To handle this semantic gap, we designed a mapping from our fine-grained labels to the target labels (detailed in Tab. 11) and employed a multi-label prediction strategy. For each audio clip, we apply a softmax function across all mapped fine-grained emotions (e.g., 38 for EmoDB). A coarse label is predicted as 'present' if the probability of any of its constituent emotions surpasses a dynamic threshold set at 1.5 times uniform chance.

Table 7: Sim-to-real generalization of EMPATHICINSIGHT-VOICE LARGE on real-world human-acted datasets EmoDB and RAVDESS. Accuracies are reported per emotion category.

| Emotion | EmoDB | RAVDESS |
|---|---|---|
| Anger | 95.3% | 88.5% |
| Boredom | 28.4% | – |
| Disgust | 28.3% | 33.9% |
| Fear | 62.3% | 92.2% |
| Happiness | 74.7% | 92.2% |
| Neutral | 100.0% | 100.0% |
| Sadness | 74.2% | 48.4% |
| Surprise | – | 97.9% |
| Calm | – | 53.1% |
| **Overall** | **70.6%** | **74.2%** |

The results, shown in Tab. 7, demonstrate strong generalization. Our model achieved an overall accuracy of 70.6% on EmoDB and 74.2% on RAVDESS. Performance is particularly high for high-arousal emotions with clear acoustic signatures, such as Anger (95.3% on EmoDB) and Surprise (97.9% on RAVDESS). Performance on categories like Boredom and Disgust is lower, which may reflect both the inherent subtlety of these emotions and potential mismatches in (overly) acted portrayal between datasets. Despite the inherent noise and subjectivity of mapping labels, these strong above-chance results validate that EMONET-VOICE BIG enables the learning of robust SER representations that successfully transfer to human speech.

## 5 DISCUSSION

**ASR models don't (yet) understand emotions?** ASR models like Whisper currently lack the ability to accurately understand and represent nuanced emotions (Dutta & Ganapathy, 2024; Li et al., 2023). However, our work shows that with continually pretraining, ASR models can begin to perceive and respond to emotional cues in ways that support more human-like predictions. Our EMONET-VOICE BIG dataset represents a crucial first step toward equipping AI models with this emotional understanding. In particular, our real-to-sim generalization results offer promising evidence: models trained on synthetic data learn representations that transfer well to human speech, achieving strong results on established benchmarks. While a sim-to-real gap remains, our findings suggest it is surmountable, emphasizing synthetic data as a viable foundation for training robust SER systems.

**Arousal-Dependent Recognition Bias.** At the same time, our fine-grained evaluation on EMONET-VOICE BENCH reveals the profound challenges that remain. We observe a clear hierarchy where high-arousal, acoustically salient emotions like *Anger* are well-recognized, while subtle, low-arousal states like *Concentration* or *Contentment* remain elusive for all tested models. This suggests that current architectures may be overly reliant on simple prosodic cues (pitch, energy) and struggle with the nuanced signatures of internal states.

**Annotation Ambiguity Predicts Model Performance.** Perhaps the most crucial insight comes from the relationship between human annotator agreement and model performance. Emotions with high expert consensus consistently yield high model performance, while those with low agreement lead to near-chance model accuracy. This pattern suggests that inter-annotator agreement may represent a practical upper bound on performance for subjective tasks like SER. Furthermore, our analysis on distinguishing *Sadness* from *Distress* shows that our benchmark succeeds in its primary goal: to quantify the difficulty of nuanced emotional distinctions. Rather than being a failure of the model, the modest 63% accuracy is a success of the benchmark in providing a concrete metric for a problem that was previously difficult to even define. It moves the field from simple classification toward measuring the resolution of a model's emotional understanding.

**The Cognitive Emotion Recognition Gap.** A particularly noteworthy pattern emerges for cognitively-oriented emotions—states that require contextual understanding beyond immediate acoustic features. Emotions such as Contemplation, Interest, and Concentration represent mental processes rather than affective responses, and their recognition may fundamentally require understanding *why* someone is in a particular state, not merely *how* they sound while experiencing it. This limitation points to a broader challenge in current emotion recognition paradigms: the reliance on acoustic features alone may be insufficient for detecting emotions that are primarily cognitive rather than affective. Future architectures might need to incorporate contextual information, dialogue history, or multimodal inputs to bridge this gap, going toward multimodal AI assistants.

**Limitations.** While EMONET-VOICE represents a significant step forward in SER, several limitations point to important directions for future work. First, our prompting approach involves actors simulating emotions. This was necessary to generate clear, labeled instances across 40 emotions, but such performances differ from spontaneous, real-world emotional speech, which tends to be more nuanced and blended. Despite strong generalization results, bridging the gap between our benchmark and natural conversational speech remains a core challenge. Second, although EMONET-VOICE includes 11 distinct voices across 4 languages—more diverse than prior work—it does not fully represent the global range of accents, dialects, age groups, or cultural vocal styles. However, the synthetic pipeline is designed to be easily expanded to improve this coverage over time. Third, all data was generated using GPT-4o Audio. While this offers consistency and control, it may also introduce model-specific acoustic artifacts or biases. Mitigating such single-source bias through multi-model data generation is a priority for future iterations. Lastly, emotion perception is inherently subjective (Schuhmann et al., 2025). Our high-agreement labels with intensities, derived from unanimous expert consensus, offer a reliable benchmark—but they reflect only one interpretation. To support broader research, we also release lower-agreement samples that capture the ambiguity and complexity of emotional expression.

## 6 CONCLUSION

We introduced EMONET-VOICE, a suite of novel datasets for fine-grained SER, designed to overcome critical limitations of existing SER resources. This includes EMONET-VOICE BIG, a large-scale synthetic multilingual pretraining dataset, and EMONET-VOICE BENCH, an expert-annotated benchmark covering 40 emotion categories with 3-level ratings. Their synthetic design ensures privacy, diversity, and scalability. Furthermore, we also release EMPATHICINSIGHT-VOICE models (Small and Large), which set a new standard, outperforming foundation models such as Gemini, GPT-4o, and Hume.

Our findings reveal persistent gaps in SER and highlight several research directions: examining agreement-performance dynamics across modalities (text, video, physiological signals), building targeted architectures for low-agreement categories, and developing context-aware models to bridge cognitive recognition challenges. Expanding EMONET-VOICE with more samples, languages, and speakers, incorporating multiple generative models, and probing cross-modal consistency (e.g., linking speech with facial expressions) present promising paths for richer benchmarks and models.

ETHICS STATEMENT

This work addresses concerns about unintended effects of emotionally uncalibrated AI. As AI models become more capable of producing emotionally charged content, it is essential to understand how people interpret and respond to these synthetic expressions. Our datasets enable the study of risks such as miscommunication and manipulation, underscoring the need for safeguards (Helff et al., 2025). The development of EMONET-VOICE was guided by a strong ethical commitment, primarily addressed through the exclusive use of synthetic voice generation. To minimize privacy risks, EMONET-VOICE relies exclusively on synthetic voice generation, avoiding the collection of sensitive human emotional data. While unlikely, we note the remote possibility that synthetic samples could resemble real individuals (Hintersdorf et al., 2024); however, no personally identifiable data was used at any stage. We applied prompt diversification to reflect a broad range of gender, demographic, and accent representations while minimizing problematic content, motivated by Friedrich et al. (2025). We release EMONET-VOICE as a research artifact intended for academic use.

REPRODUCIBILITY STATEMENT

To ensure reproducibility, we will release all code for data generation, model training, and evaluation. We provide all resources in the supplement. The EMONET-VOICE BIG and EMONET-VOICE BENCH datasets are publicly available on Hugging Face [link will be provided upon acceptance]. Our trained EMPATHICINSIGHT-VOICE models will also be released with instructions for inference. The expert annotation protocol and the full label mappings used for cross-dataset evaluation are detailed in the appendix, providing all necessary information for others to replicate our findings.

LLM USAGE

We used Large Language Models (LLMs) in several capacities during this research. GPT-4 was used to assist in the initial extraction of emotion concepts from literature during the taxonomy construction phase, as described in Appendix A.4. Gemini Flash 2.0 was used for the large-scale, automated annotation of our pre-training data, as detailed in Section 3.4. Finally, we used an LLM for assistance with grammar, clarity, and rephrasing during the writing of this manuscript. All final claims, data, and written text were reviewed and verified by the human authors, who take full responsibility for the content of this paper.

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

# A  APPENDICES

## A.1  EMONET-VOICE TAXONOMY

The 40 emotion categories used in EMONET-VOICE, adapted from EMONET-FACE (Schuhmann et al., 2025), are listed below with associated descriptive terms used during conceptualization and prompting:

- **Amusement:** 'lighthearted fun', 'amusement', 'mirth', 'joviality', 'laughter', 'playfulness', 'silliness', 'jesting'
- **Elation:** 'happiness', 'excitement', 'joy', 'exhilaration', 'delight', 'jubilation', 'bliss', 'Cheerfulness'
- **Pleasure/Ecstasy:** 'ecstasy', 'pleasure', 'bliss', 'rapture', 'Beatitude'
- **Contentment:** 'contentment', 'relaxation', 'peacefulness', 'calmness', 'satisfaction', 'Ease', 'Serenity', 'fulfillment', 'gladness', 'lightness', 'serenity', 'tranquility'
- **Thankfulness/Gratitude:** 'thankfulness', 'gratitude', 'appreciation', 'gratefulness'
- **Affection:** 'sympathy', 'compassion', 'warmth', 'trust', 'caring', 'Clemency', 'forgiveness', 'Devotion', 'Tenderness', 'Reverence'
- **Infatuation:** 'infatuation', 'having a crush', 'romantic desire', 'fondness', 'butterflies in the stomach', 'adoration'
- **Hope/Enthusiasm/Optimism:** 'hope', 'enthusiasm', 'optimism', 'Anticipation', 'Courage', 'Encouragement', 'Zeal', 'fervor', 'inspiration', 'Determination'
- **Triumph:** 'triumph', 'superiority'
- **Pride:** 'pride', 'dignity', 'self-confidently', 'honor', 'self-consciousness'
- **Interest:** 'interest', 'fascination', 'curiosity', 'intrigue'
- **Awe:** 'awe', 'awestruck', 'wonder'
- **Astonishment/Surprise:** 'astonishment', 'surprise', 'amazement', 'shock', 'startlement'
- **Concentration:** 'concentration', 'deep focus', 'engrossment', 'absorption', 'attention'
- **Contemplation:** 'contemplation', 'thoughtfulness', 'pondering', 'reflection', 'meditation', 'Brooding', 'Pensiveness'
- **Relief:** 'relief', 'respite', 'alleviation', 'solace', 'comfort', 'liberation'
- **Longing:** 'yearning', 'longing', 'pining', 'wistfulness', 'nostalgia', 'Craving', 'desire', 'Envy', 'homesickness', 'saudade'
- **Teasing:** 'teasing', 'bantering', 'mocking playfully', 'ribbing', 'provoking lightly'
- **Impatience and Irritability:** 'impatience', 'irritability', 'irritation', 'restlessness', 'short-temperedness', 'exasperation'
- **Sexual Lust:** 'sexual lust', 'carnal desire', 'lust', 'feeling horny', 'feeling turned on'
- **Doubt:** 'doubt', 'distrust', 'suspicion', 'skepticism', 'uncertainty', 'Pessimism'
- **Fear:** 'fear', 'terror', 'dread', 'apprehension', 'alarm', 'horror', 'panic', 'nervousness'
- **Distress:** 'worry', 'anxiety', 'unease', 'anguish', 'trepidation', 'Concern', 'Upset', 'pessimism', 'foreboding'
- **Confusion:** 'confusion', 'bewilderment', 'flabbergasted', 'disorientation', 'Perplexity'
- **Embarrassment:** 'embarrassment', 'shyness', 'mortification', 'discomfiture', 'awkwardness', 'Self-Consciousness'
- **Shame:** 'shame', 'guilt', 'remorse', 'humiliation', 'contrition'
- **Disappointment:** 'disappointment', 'regret', 'dismay', 'letdown', 'chagrin'
- **Sadness:** 'sadness', 'sorrow', 'grief', 'melancholy', 'Dejection', 'Despair', 'Self-Pity', 'Sullenness', 'heartache', 'mournfulness', 'misery'
- **Bitterness:** 'resentment', 'acrimony', 'bitterness', 'cynicism', 'rancor'
- **Contempt:** 'contempt', 'disapproval', 'scorn', 'disdain', 'loathing', 'Detestation'
- **Disgust:** 'disgust', 'revulsion', 'repulsion', 'abhorrence', 'loathing'
- **Anger:** 'anger', 'rage', 'fury', 'hate', 'irascibility', 'enragement', 'Vexation', 'Wrath', 'Peevishness', 'Annoyance'
- **Malevolence/Malice:** 'spite', 'sadism', 'malevolence', 'malice', 'desire to harm', 'schadenfreude'
- **Sourness:** 'sourness', 'tartness', 'acidity', 'acerbity', 'sharpness' (Note: Primarily gustatory, vocal correlates might be subtle reactions)
- **Pain:** 'physical pain', 'suffering', 'torment', 'ache', 'agony'
- **Helplessness:** 'helplessness', 'powerlessness', 'desperation', 'submission'

- **Fatigue/Exhaustion:** 'fatigue', 'exhaustion', 'weariness', 'lethargy', 'burnout', 'Weariness'
- **Emotional Numbness:** 'numbness', 'detachment', 'insensitivity', 'emotional blunting', 'apathy', 'existential void', 'boredom', 'stoicism', 'indifference'
- **Intoxication/Altered States of Consciousness:** 'being drunk', 'stupor', 'intoxication', 'disorientation', 'altered perception'
- **Jealousy & Envy:** 'jealousy', 'envy', 'covetousness'

A.2   MORE DETAILS ON SOTA SER MODEL TRAINING METHODOLOGY

This section provides an in-depth description of the training procedures for the models discussed in Section 4: i.e. the Whisper backbone and the EMPATHICINSIGHT-VOICE ensembles.

**Data Curation and Fine-tuning for Emotion Captioning.**   Our goal was to adapt pre-trained Whisper models (Radford et al., 2023) for the task of generating nuanced emotional captions from speech. The data generation and fine-tuning pipeline involved several key steps:

1. **Initial Large-Scale Data Sources:** The primary data source was the EMONET-VOICE BIG synthetic voice-acting dataset. This was augmented with approximately 4,500 hours of audio extracted from publicly available online videos (vlogs, diaries, documentaries). We applied voice activity detection (VAD) to isolate speech segments ranging from 3 to 12 seconds.

2. **Dimensional Emotion Scoring with Gemini Flash 2.0:** All audio snippets—both from EMONET-VOICE BIG and the VAD-extracted clips—were annotated using Gemini Flash 2.0. A complex, multi-shot prompt (detailed in the supplementary materials) guided the model to produce intensity scores on a 0–4 scale (0 = absent, 4 = extremely present) for each of our 40 emotion dimensions simultaneously. This provided a structured, dimensional representation of perceived emotional content.

3. **Iterative Caption Generation for Whisper Training:**

   - Our initial attempt was to fine-tune Whisper to *directly regress* these 40-dimensional scores (i.e., to output numerical values), but this approach consistently collapsed into predicting nonsensical sequences of numbers. Similarly, training a specialized output head to perform ordinal regression utilizing a Wasserstein distance loss did not yield more sophisticated or coherent captions.

   - We then converted the dimensional scores into *procedurally generated string captions* using predefined templates (e.g., "The speaker sounds strongly amused and slightly joyful."). Training on these templated captions improved over direct regression, but the resulting Whisper outputs still tended toward repetitive or syntactically unnatural phrasing.

   - The most effective strategy was to take those procedurally generated captions and run them back through Gemini Flash 2.0 for *paraphrasing*. This second pass introduced significant linguistic diversity and more natural sentence structures, while preserving the original 40-dimensional semantics. The paraphrasing prompt specifically encouraged varied wording and sentence complexity.

4. **Training Data Preparation:** All EMONET-VOICE BIG audio segments longer than 30 seconds were truncated to their first 30 seconds, to meet Whisper's input constraints. Very long segments were further subdivided at silent regions into shorter clips, resulting in a final training pool of over 2 million audio–caption pairs when combined with the processed VAD data.

5. **Whisper Fine-tuning:** Various sizes of OpenAI's Whisper models were then fine-tuned on this dataset of audio paired with the paraphrased emotional captions. The objective was to teach Whisper to generate fluid, context-sensitive descriptions of emotional content given raw speech input. Iteratively refining the captions via paraphrasing proved crucial for yielding outputs that were both semantically accurate and linguistically natural. We also experimented with incorporating synthetic "emotion bursts" during fine-tuning, but this led to degraded embedding quality and was therefore not used in the final models.

**EMPATHICINSIGHT-VOICE: MLP Ensembles for Dimensional Emotion Prediction.** The EMPATHICINSIGHT-VOICE models were designed to provide direct predictions for each of the 40 emotion dimensions—complementing the captioning approach with explicit scalar estimates.

1. **Feature Extraction:** We used the encoder from our best-performing Whisper variant as a fixed feature extractor. For any input audio, we ran it through the Whisper encoder and collected the full sequence of token embeddings (sequence length = 1,500; embedding dimension = 768), yielding 1,152,000 features when flattened. Preliminary experiments showed that preserving the entire unpooled sequence outperformed all tested pooling strategies (mean, max, min, concatenation) for downstream MLP regression.

2. **MLP "Expert" Heads:** We trained an ensemble of 40 independent MLP models. Each MLP served as an "expert" head dedicated to regressing the intensity score for exactly one of the 40 emotion dimensions using the corresponding flattened Whisper embeddings as input.

3. **Training Targets:** The regression targets were the direct 0–4 intensity scores produced by Gemini Flash 2.0 (via the multi-shot prompt described in the supplementary files). During the *encoder fine-tuning* stage, we experimented with injecting synthetic "emotion bursts"—artificially boosting certain dimension signals in the audio—to encourage a more robust embedding space. However, this augmentation degraded the underlying Whisper embeddings and ultimately hurt downstream MLP performance. Consequently, no synthetic bursts were used for final training.

4. **MLP Architecture:** Both the Small and Large EMPATHICINSIGHT-VOICE variants share the same overall architectural pattern for regressing from the high-dimensional flattened embeddings:

   - *Input Projection:* A first linear layer reduces the 1,152,000-dimensional input to a much smaller embedding space.
   - *Hidden Layers:* Three fully connected layers with ReLU activations, each followed by dropout for regularization to mitigate overfitting.
   - *Output Layer:* A final linear projection that outputs a single continuous value in [0, 4], corresponding to the predicted intensity for that emotion.

5. **Model Sizes:**

   - EMPATHICINSIGHT-VOICE SMALL: The initial projection reduces 1,152,000 inputs to 64 dimensions. The subsequent hidden layer sizes are $64 \rightarrow 32 \rightarrow 16$. Each MLP head has about 73.73 million trainable parameters, the vast majority residing in that first projection layer.
   - EMPATHICINSIGHT-VOICE LARGE: The initial projection reduces 1,152,000 inputs to 128 dimensions. The subsequent hidden layers are $128 \rightarrow 64 \rightarrow 32$. This yields approximately 147.48 million trainable parameters per head, again dominated by the input projection.

6. **Parallel Inference and Training Loss:** At inference time, we evaluate all 40 MLP experts in parallel to predict the full 40-dimensional emotion profile (i.e., different strengths of emotionality across dimensions). During training, each MLP head is optimized independently using the mean absolute error (MAE) between predicted and target emotion strength.

All trained EMPATHICINSIGHT-VOICE models (Small and Large) and the associated inference code are available via our project page.

### A.3 HUME VOICE MAPPING

### A.4 DETAILED TAXONOMY CONSTRUCTION METHODOLOGY

The 40-category emotion taxonomy utilized in both the EMONET-VOICE foundation and benchmark datasets was originally developed for the EmoNet-Face Benchmark (Schuhmann et al., 2025).

The primary objective was to create a taxonomy that supports a more fine-grained and nuanced understanding of affective states in AI, moving beyond the limitations of traditional basic emotion

| Hume Voice Label | Our Taxonomy |
|---|---|
| Joy | Elation |
| Empathic Pain | Distress |
| Guilt | - |
| Nostalgia | Longing |
| Determination | - |
| Surprise (positive) | Surprise |
| Horror | Fear |
| Calmness | Contentment |
| Desire | Sexual Lust |
| Awkwardness | Embarrassment |
| Satisfaction | Pleasure |
| Aesthetic Appreciation | Awe |
| Entrancement | Concentration |
| Romance | Infatuation |
| Love | Affection |
| Excitement | Arousal |
| Realization | Contemplation |
| Tiredness | Fatigue |
| Envy | Jealousy & Envy |
| Anxiety | - |
| Boredom | - |
| Adoration | - |
| Sympathy | - |
| Admiration | Admiration |
| Craving | Craving |
| Surprise (negative) | Astonishment |

Table 8: Mapping of Hume Voice labels to our emotion taxonomy. Note that if one Hume Voice label fits to more than one emotion from our taxonomy, only one item was chosen.

Table 9: Summary of key dataset statistics for EMONET-VOICE. *Hume Voice provides 46 emotions on a continuous scale from 0-1, of which we were able to map 29 to our emotion taxonomy. Human annotators voted on a discrete scale: 0 (emotion not present), 1 (emotion weakly present), 2 (emotion strongly present). All scales were transformed to a 0-10 scale for further analysis. Note that GPT-4o Audio Preview was not able to process 2,100 samples (e.g., returned an empty response).

| Annotator | Unique Audio Files | Emotions per Annotation | Scale |
|---|---|---|---|
| Human 1 | 6837 | 1 | 0-2 |
| Human 2 | 6620 | 1 | 0-2 |
| Human 3 | 2600 | 1 | 0-2 |
| Human 4 | 11605 | 1 | 0-2 |
| Human 5 | 343 | 1 | 0-2 |
| Human 6 | 5600 | 1 | 0-2 |
| EMPATHICINSIGHT-VOICE LARGE | 12600 | 40 | 0-4 |
| EMPATHICINSIGHT-VOICE SMALL | 12600 | 40 | 0-4 |
| GPT-4o Audio Preview 2024-12-17 | 10500 | 40 | 0-10 |
| GPT-4o Mini Audio Preview | 12600 | 40 | 0-10 |
| Gemini 2.0 Flash | 12600 | 40 | 0-10 |
| Gemini 2.5 Pro | 12600 | 40 | 0-10 |
| Hume Voice | 12600 | *29 | 0-1 |

models. This development was rooted in contemporary psychological research and significantly informed by the principles of the Theory of Constructed Emotion (TCE) (Barrett, 2017).

The taxonomy was designed to encompass a wide array of affective experiences, including not only common positive and negative emotions but also intricate social emotions (e.g., *Embarrassment*,

| emotion | alpha | alpha ci lower | alpha ci upper | n items |
|---|---|---|---|---|
| Embarrassment | 0.272 | 0.186 | 0.368 | 300 |
| Teasing | 0.271 | 0.178 | 0.362 | 300 |
| Pain | 0.247 | 0.160 | 0.334 | 300 |
| Anger | 0.220 | 0.129 | 0.310 | 300 |
| Shame | 0.216 | 0.122 | 0.297 | 300 |
| Sadness | 0.211 | 0.111 | 0.301 | 300 |
| Distress | 0.208 | 0.121 | 0.297 | 300 |
| Malevolence | 0.204 | 0.098 | 0.294 | 300 |
| Contentment | 0.197 | 0.109 | 0.281 | 300 |
| Relief | 0.196 | 0.098 | 0.280 | 300 |
| Jealousy / Envy | 0.194 | 0.095 | 0.282 | 300 |
| Intoxication | 0.193 | 0.104 | 0.279 | 300 |
| *Authenticity* | 0.185 | 0.093 | 0.279 | 300 |
| Disappointment | 0.176 | 0.071 | 0.271 | 300 |
| Fear | 0.161 | 0.066 | 0.241 | 300 |
| Impatience and Irritability | 0.159 | 0.057 | 0.247 | 300 |
| Helplessness | 0.158 | 0.070 | 0.246 | 300 |
| Pride | 0.156 | 0.057 | 0.241 | 300 |
| Sexual Lust | 0.149 | 0.048 | 0.243 | 300 |
| Triumph | 0.145 | 0.043 | 0.246 | 300 |
| Elation | 0.138 | 0.040 | 0.229 | 300 |
| **Overall** | 0.138 | 0.124 | 0.152 | 12600 |
| Fatigue | 0.129 | 0.034 | 0.217 | 300 |
| Concentration | 0.103 | 0.023 | 0.186 | 300 |
| Disgust | 0.103 | 0.004 | 0.195 | 300 |
| Thankfulness | 0.088 | -0.008 | 0.178 | 300 |
| Pleasure | 0.082 | -0.011 | 0.177 | 300 |
| Doubt | 0.078 | -0.020 | 0.171 | 300 |
| Amusement | 0.068 | -0.031 | 0.155 | 300 |
| Infatuation | 0.063 | -0.031 | 0.150 | 300 |
| Confusion | 0.060 | -0.027 | 0.148 | 300 |
| Contempt | 0.046 | -0.045 | 0.130 | 300 |
| Affection | 0.043 | -0.052 | 0.133 | 300 |
| Bitterness | 0.033 | -0.051 | 0.116 | 300 |
| Astonishment | 0.021 | -0.068 | 0.109 | 300 |
| Contemplation | 0.021 | -0.065 | 0.104 | 300 |
| Sourness | 0.004 | -0.088 | 0.083 | 300 |
| Hope | -0.005 | -0.106 | 0.090 | 300 |
| Longing | -0.046 | -0.149 | 0.046 | 300 |
| *Arousal* | -0.066 | -0.170 | 0.030 | 300 |
| Interest | -0.094 | -0.178 | -0.009 | 300 |
| Emotional Numbness | -0.099 | -0.179 | -0.017 | 300 |
| Awe | -0.127 | -0.218 | -0.035 | 300 |

Table 10: Cronbach's $\alpha$ inter-rater reliability (0 = emotion absent; 1 = weakly present; 2 = strongly present) for each emotion category ($n = 300$ items per label), with 95% confidence intervals obtained via non-parametric bootstrap (1 000 resamples, seed = 42). "Overall" reports $\alpha$ and CI computed across all 40 emotion categories + 2 extra categories (12 000 + 600 = 12 600 total annotations). Note that the analysis contains two extra categories (authenticity and arousal) that is not present in the narrow emotion category definition A.1.

*Shame*, *Pride*), cognitive states (e.g., *Concentration*, *Doubt*, *Confusion*), and bodily states (e.g., *Pain*, *Fatigue*, *Intoxication*). Less typical but experientially relevant categories like *Sourness* and *Helplessness* were also incorporated. The full list of 40 categories and their descriptive word clusters can be found in App. A.1 (cross-referencing the list you already have, which is similar to App. Tab. 4 from the EmoNet-Face paper).

**Audio Emotion Annotation**

**Instructions**

In this task, you'll be assessing whether a specific emotion appears to be present in the audio recordings. Each recording will be associated with a single emotion label, and you need to decide whether that emotion is:

- **Not Present** – The emotion is not detectable in the audio

- **Weakly Present** – The emotion is somewhat present but not strong

- **Strongly Present** – The emotion is clearly and strongly expressed

Listen carefully to each recording and make your selection based on your perception of the emotion in the audio.

Figure 2: Instructions given to the human annotator for the expert annotation of EMONET-VOICE BENCH.

The construction process involved several key stages:

1. **Literature-Driven Candidate Extraction:** The comprehensive "Handbook of Emotions" (946 pages) (Lewis et al., 2016) was digitized using Optical Character Recognition (OCR). The digitized text was then divided into manageable 500-word segments.

2. **AI-Assisted Term Identification:** GPT-4 was employed to analyze these text segments and extract potential nouns representing emotion concepts.

3. **Refinement and Deduplication:** The initially extracted terms were aggregated, and duplicates were removed, resulting in a candidate list of approximately 170 unique emotion-related nouns.

4. **Expert-Guided Clustering and Categorization:** This refined list of terms underwent an iterative process of clustering. This involved independent categorization efforts by team members, followed by critical reviews and discussions. Psychologists and researchers in affective computing provided expert guidance throughout this phase to ensure the semantic coherence and psychological relevance of the emerging categories. Each of the final 40 categories represents a cluster of these semantically related emotion words.

In line with the Theory of Constructed Emotion, this taxonomy does not presuppose the biological universality or fixedness of these emotional categories. Instead, it is intended to facilitate context-aware and socially informed interpretations of affective expressions by AI systems. Recognizing the inherent ambiguity in perceiving emotions (e.g., a high-arousal vocal expression might be interpreted as amusement, elation, or excitement depending on context and observer), the taxonomy was specifically designed to support plausible multi-label annotations rather than forcing rigid, single-label classifications. This approach aims to enable richer and more contextually sensitive representations of emotion in AI.

## B  ANNOTATION PLATFORM INSTRUCTIONS AND UI

## C  FINE-GRAINED TO COARSE EMOTION MAPPING

Table 11 details the mappings used for our cross-dataset evaluation. Bridging the semantic gap between our 40 fine-grained categories and the 7-8 coarse labels of EmoDB and RAVDESS is a non-trivial challenge. The following mappings represent a best-effort, yet inherently subjective, alignment.

The differences between the two mappings arise because the target taxonomies are distinct; for instance, EmoDB includes 'boredom' while RAVDESS has 'calm' and 'surprised'. This subjectivity

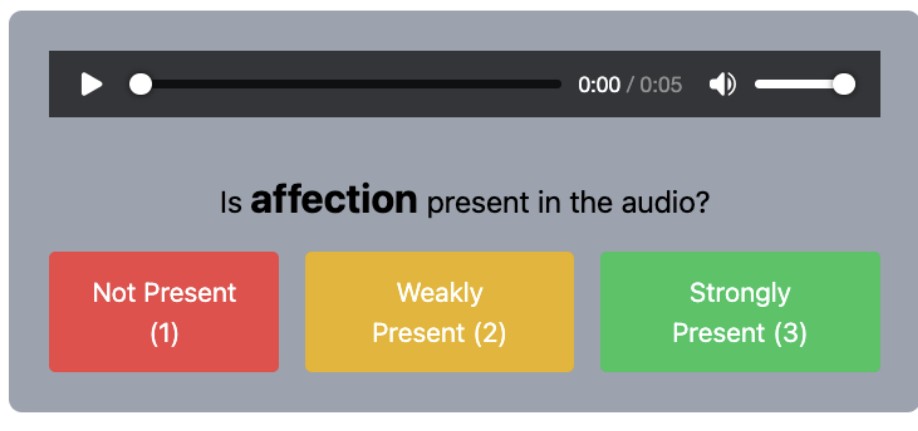

Figure 3: UI of our expert annotation tool for EMONET-VOICE BENCH.

is important context for our results: we acknowledge that lower performance on certain categories (e.g., 'boredom' or 'disgust') may reflect a poor semantic fit in the mapping rather than a fundamental model deficiency. A '—' indicates an unmapped emotion.

Table 11: Mapping from EMONET-VOICE's 40 fine-grained emotion categories to the coarse categories of EmoDB ($n = 7$: **anger**, *boredom*, **disgust**, **fear**, **happiness**, **sadness**, **neutral**) and RAVDESS ($n = 8$: **angry**, *calm*, **disgust**, **fearful**, **happy**, **neutral**, **sad**, *surprised*). Shared categories are in **bold**; unique in *italics*.

| EMONET-VOICE Fine-Grained Emotion | EmoDB Mapping | RAVDESS Mapping |
|---|---|---|
| Affection | happiness | calm |
| Amusement | happiness | happy |
| Anger | anger | angry |
| Astonishment/Surprise | fear | surprised |
| Awe | — | calm |
| Bitterness | anger | angry |
| Concentration | neutral | neutral |
| Contemplation | neutral | calm |
| Contempt | disgust | disgust |
| Contentment | happiness | happy |
| Disappointment | sadness | sad |
| Disgust | disgust | disgust |
| Distress | fear | fearful |
| Doubt | neutral | surprised |
| Elation | happiness | happy |
| Embarrassment | sadness | sad |
| Emotional Numbness | boredom | neutral |
| Fatigue/Exhaustion | sadness | disgust |
| Fear | fear | fearful |
| Helplessness | sadness | fearful |
| Hope/Enthusiasm/Optimism | happiness | happy |
| Impatience and Irritability | anger | angry |
| Infatuation | happiness | happy |
| Interest | neutral | surprised |
| Intoxication/Altered States | — | calm |
| Jealousy / Envy | sadness | angry |
| Longing | sadness | sad |
| Malevolence/Malice | anger | angry |
| Pain | sadness | sad |
| Pleasure/Ecstasy | happiness | happy |
| Pride | happiness | happy |
| Relief | happiness | calm |
| Sadness | sadness | sad |
| Sexual Lust | — | — |
| Shame | sadness | disgust |
| Sourness | disgust | disgust |
| Teasing | happiness | happy |
| Thankfulness/Gratitude | happiness | happy |
| Triumph | happiness | happy |
| Confusion | neutral | surprised |