# OpenReview forum: "EmoNet-Voice: A Large-Scale Synthetic Benchmark for Fine-Grained Speech Emotion"
_ICLR.cc/2026/Conference — ICLR 2026 Conference Withdrawn Submission_

### Official Review · Reviewer_AcgU · 2025-10-22

**Soundness:** 1
**Presentation:** 1
**Contribution:** 2
**Rating:** 4
**Confidence:** 4

**Summary:**

This paper proposes a large-scale, synthetic, multilingual benchmark for speech emotion recognition, which comprises two components:
- EmoNet-Voice Big, a 5,000-hour fully synthetic dataset covering 40 emotion categories across 11 voices and 4 languages, generated via GPT-4o.
- EmoNet-Voice Bench, a 35.8h benchmark annotated by psychology experts with 3-level intensity ratings.

This paper also proposes Empathicinsight-Voice, models based on continually pre-trained Whisper encoders with MLP heads for 40 emotion regressions.

**Strengths:**

- EmoNet-Voice Big includes 40 fine-grained emotion categories, extending beyond conventional basic emotion datasets that typically cover fewer than ten classes.

**Weaknesses:**

- The entire dataset is generated using GPT-4o Audio, which raises concerns about the acoustic realism and variability of emotional expressions. Although the authors argue that synthetic data enable ethical and scalable emotion modeling, it remains unclear whether these voices exhibit natural prosodic patterns comparable to genuine human affect. The reported sim-to-real transfer results may primarily reflect similarities between acted emotional speech and synthetic data rather than genuine emotion understanding, which is not sufficient to be convincing (at least for me).

- The statement that the model “hears what humans hear” is somewhat overstated; a Pearson correlation around 0.42 cannot substantiate that level of human alignment.

- The claim that the authors “further contribute by developing novel state-of-the-art SER models” appears overstated, since the training data are entirely generated by GPT-4o Audio, effectively distilling GPT-4o’s existing expressive or affective capabilities rather than proposing a new modeling approach. In fact, many companies have already adopted similar distilling methods, making it neither novel nor of genuine scholarly significance.

- No demo page or audio examples are provided, which prevents readers from verifying the claimed perceptual quality and emotional realism of the synthetic voices.

**Questions:**

- Some figures and tables (e.g., Figure 1, Table 6) are visually overwhelming, which may reduce readability and make it difficult to extract key insights.

- The writing quality has several issues, including inconsistent quotation mark usage (mixing single and double quotes, and lacking proper left–right distinction), as well as occasional rhetorical and informal phrasing that does not align with standard academic writing conventions.

---

### Official Review · Reviewer_cSWm · 2025-10-29

**Soundness:** 3
**Presentation:** 3
**Contribution:** 1
**Rating:** 2
**Confidence:** 4

**Summary:**

This article proposes a new synthetic speech emotional dataset (EMONET-VOICE BIG) produced by GPT-4 OmniAudio, containing 40 emotions in 4 languages with a variety of accents, for a total of 4752 hours of audio.
Then, a subset of 12,600 clips from the dataset are annotated by 2 to 4 human annotators, giving a validated dataset of 36,26 hours (EMONET-VOICE BENCH).
A Whisper encoder is trained on the EMONET-VOICE BIG set, frozen, then the classification heads for each emotion are trained on EMONET-VOICE BENCH, giving a SER model labelled as EMPATHIC INSIGHT-VOICE.
This SER model is compared to a set of generalist commercial audio LLMs by their correlation with human predictions, and the accuracy per emotion is measured on the EmoDB and RAVDESS acted datasets.

**Strengths:**

The article is well structured, with many details about the datasets contents and creation.
The idea of integrating synthetic emotional speech in training datasets is not new, but training a model from scratch on non-validated synthetic data is interesting.

**Weaknesses:**

The bibliography lacks an important set of datasets: the speech emotion recognition datasets recorded in the while and annotated a posteriori, such as MSP-Podcast or MSP-conversations.
Only the EMONET-VOICE BENCH dataset has been validated by humans, which makes the use of the EMONET-VOICE BIG set limited, as we don't have qualitative measures about the quality of the generated emotional speech.
The annotation by up to 4 distinct annotators, and mostly by only 2 annotators limits the value of the annotation as well, which would be first reason why Cronbach’s α is so low.

If the article manages to create a SER model that is efficient at characterizing "how is the synthetic speech from AudioLLM perceived by humans", the transfer to real natural emotional speech has not been demonstrated, which restrict the range of the article.
The only test done on real speech is on acted datasets, RAVDESS and EmoDB, and the lack of comparison with any baseline SER model is limited.

Small typo line 147: is 'Contempt' a negative emotion?

**Questions:**

The Whisper model has been known as one of the foundational models for SER in the previous years, with HuBERT/Wav2Vec2.0/WavLM models, however, it is often associated with the extraction of emotional content from the linguistic, while the acoustic aspect is often taken care by different models. As the dataset is generated from textual prompts, do you think there could be a bias here, where most of the emotional expression is coming from the linguistic content rather than the prosody?

---

### Official Review · Reviewer_rjSj · 2025-10-31

**Soundness:** 2
**Presentation:** 3
**Contribution:** 2
**Rating:** 4
**Confidence:** 4

**Summary:**

The paper proposes EMONET-VOICE, a collection of datasets and models for speech emotion understanding. In particular, it introduces (i) EmoNet-Voice Big a ~5k h multilingual synthetic dataset covering 40 fine-grained emotion categories, and (ii) EmoNet-Voice Bench, a 4.7k sample benchmark annotated by psychology experts with 3-level intensity ratings. Using these datasets, the authors train a family of speech emotion recognition models (Empathic Insight Voice Small/Large) based on continually pre-trained Whisper encoders and per-emotion expert heads. The models show promising results in aligning with human ratings and demonstrate some degree of sim-to-real generalization to existing emotion datasets such as EmoDB and RAVDESS.

**Strengths:**

1. Fine-grained taxonomy (40 emotions) and intensity-aware evaluation.
2. Large-scale synthetic multilingual dataset.

**Weaknesses:**

1. Data are completely synthetic and do not represent naturalistic settings. In addition, they are all generated by GPT-4o, which can introduce bias concerns specific to that model’s generations.
2. Low inter-rater reliability (α~0.14 overall).
3. Training on Gemini-generated targets without quantifying Gemini–human agreement.

**Questions:**

1. Since your model is trained on Gemini-generated targets, what is the correlation between Gemini’s outputs and human ratings?
2. Why do you optimize the model with mean absolute error? Did you try training with a binary classification objective instead? It seems that you already have labels for whether each emotion is present or not.
3. In Sec. 3.3 you mention that in cases where two annotators agree, you bring in a third one. How do you handle the samples where annotations don’t agree?
4. What is the noise ceiling of your ratings, and how does the correlation results change if you account for that?
5. It is surprising that your model achieves higher correlation with Gemini 2.5 Pro than with Gemini 2.0 Flash, which you used for label generation. How do you explain this?
6. In Sec. 4.2 you claim that your model has "strong generalization" abilities on real-world data, but both evaluation datasets are recorded in controlled environments, and you do not compare with any other model.

Minor

1. Sec. 3.3: At the beginning you mention “From EMONET-VOICE BENCH” - I assume you meant “From EMONET-VOICE BIG"
2. Sec. "A.3 HUME VOICE MAPPING" is empty.

**Details Of Ethics Concerns:**

The paper uses fully synthetic speech data but represents human emotional and accent characteristics. It also involves human annotators for expert labeling

---

### Official Review · Reviewer_sS9Z · 2025-11-03

**Soundness:** 2
**Presentation:** 2
**Contribution:** 2
**Rating:** 4
**Confidence:** 3

**Summary:**

This paper introduces EMONET-VOICE BIG, a large-scale synthetic multilingual pretraining dataset, and EMONET-VOICE BENCH, a fine-grained dataset for SER. The authors release models trained on those datasets which outperform foundation models.

**Strengths:**

* All codes for data generation, model training, and evaluation will be released.Models  and datasets will be released as well.
* The paper is well-written and easy to follow, especially for the dataset construction part.

**Weaknesses:**

* The Introduction (lines 45-50) overgeneralizes the limitations of existing SER datasets. For example, MSP-Podcast is a large-scale in-the-wild dataset for SER. Besides categorical emotion labels, it also has dimensional emotion labels for sufficient granularity.
* Section 4.2 and table 7, there is no baseline for comparison. Readers cannot determine if these scores are strong or weak without comparison.
* For table 6 and 7, if the main contribution is the dataset, then you should compare the same model architecture but with different training data sources. For example, baselines should be models trained with other open-source SER datasets.

**Questions:**

Line 192, how is the subset selected to construct EMONET-VOICE BENCH?

---

### Note · Authors · 2025-11-19

**Comment:**

We sincerely thank the reviewers for their thoughtful and constructive feedback, which we will use to substantially revise and improve the manuscript. Unfortunately, given the extent of the revisions required and the limited time remaining before the deadline, we are unable to implement these changes adequately within the current review cycle. Therefore, we have decided to withdraw the paper and continue developing the work for submission to another venue. We are grateful for the time and effort invested by the reviewers and appreciate their valuable input.

**Withdrawal Confirmation:**

I have read and agree with the venue's withdrawal policy on behalf of myself and my co-authors.